# A Review of Herbal Resources Inducing Anti-Liver Metastasis Effects in Gastrointestinal Tumors via Modulation of Tumor Microenvironments in Animal Models

**DOI:** 10.3390/cancers15133415

**Published:** 2023-06-29

**Authors:** Sul-Ki Kim, Nam-Hun Lee, Chang-Gue Son

**Affiliations:** 1Liver and Immunology Research Center, Collage of Korean Medicine, Daejeon University, Daejeon 35235, Republic of Korea; zlstmfrl@naver.com; 2East-West Cancer Center, Cheonan Korean Medicine Hospital, Daejeon University, Cheonan 31099, Republic of Korea; nhlee@dju.ac.kr

**Keywords:** metastasis, cancer, tumor, microenvironments, herbs, liver, gastrointestinal tumor

## Abstract

**Simple Summary:**

Based on the importance of the tumor microenvironment and liver environment in liver metastases, we conducted a systematic review to discover herbal candidates inhibiting the liver metastasis of tumors originating from the digestive system. We produced a list of potent herbs such as *Curcuma longa Linn* and single compounds such as curcumin and tanshinone II-A, along with their underlying mechanisms. The list of herbal agents and their mechanisms produced in this study is helpful for the development of anti-liver metastasis drugs in the future via modulation of liver environments.

**Abstract:**

Liver metastases remain a major obstacle for the management of all types of tumors arising from digestive organs, and the tumor microenvironment has been regarded as an important factor in metastasis. To discover herbal candidates inhibiting the liver metastasis of tumors originating from the digestive system via the modulation of the tumor microenvironment and liver environment, we searched three representative public databases and conducted a systematic review. A total of 21 studies that employed experimental models for pancreatic (9), colon (8), and stomach cancers (4) were selected. The herbal agents included single-herb extracts (5), single compounds (12), and multiherbal decoctions (4). *Curcuma longa Linn* was most frequently studied for its anti-colon–liver metastatic effects, and its possible mechanisms involved the modulation of tumor microenvironment components such as vascular endothelial cells and immunity in both tumor tissues and circulating cells. The list of herbal agents and their mechanisms produced in this study is helpful for the development of anti-liver metastasis drugs in the future.

## 1. Introduction

Tumors in the digestive system, including the stomach, colon, rectum, and pancreas, accounted for approximately one-fifth of both new cases of cancer and cancer-related deaths in 2020 worldwide [1]. The 5-year survival rate of these gastrointestinal (GI) cancers has recently significantly increased; however, metastasis to secondary organs is a crucial event shortening the survival period [2,3]. Due to the presence of the portal vein, which brings a substantial blood flow, the liver is the most predominant site of metastasis in patients with malignancies of GI organs and occurs in almost 50~80% of tumors of the digestive tract at the late stage [4,5]. The liver is a central organ that maintains body homeostasis; therefore, liver metastasis reduces the survival rate of cancer patients. For example, patients with liver metastasis survive for only approximately one-fifth of the period of patients without liver metastasis in both colorectal and gastric cancer [6,7].

Despite advances in tumor treatments such as surgical techniques, radiotherapy, targeted therapy, and immunology therapy, the survival rates of digestive-system cancer patients with distant metastasis remain poor compared with other cancer types [8]. On the other hand, researchers have discovered that tumor metastasis is affected not only by metastasis-favoring genetic alterations of cancer cells but also by changes in the tumor microenvironment (TME) [9,10]. The TME is the location in which cancer cells communicate with adjacent stromal cells, making tumors more prone to progression and metastatic behavior [11,12]. Contrary to the genetic aberrations predominantly responsible for tumor progression, the TME plays a key role in modulating the aggressiveness, motility, dissemination, and colonization of cancer cells in distal tissues/organs [13,14]. For example, the invasion of colorectal cancer cells into liver tissue can be accelerated under metastasis-favorable microenvironments, including hepatic steatosis, inflammation, or fibrotic circumstances [15,16]. 

There are experiments showing the potential of targeting the TME to control tumor growth as well as the metastatic risk [17,18]. As pharmaceutical resources, many herbal plants such as *Panax ginseng* C.A. Mayer and *Coptis chinensis* F. have shown anti-tumor or beneficial effects in preclinical and clinical studies [19,20]. Some researchers have also reported the anti-metastatic properties of herbal remedies, especially in cancers of the GI tract, along with the effects on the TME [21,22]. These findings indicate the potential of herbal resources to inhibit the liver metastasis of GI cancers, especially given their effects on the TME.

To help develop herb-derived agents to protect against liver metastasis, we systematically searched for studies of herbal candidates inhibiting the liver metastasis of digestive organ tumors by modulating the TME.

## 2. Materials and Methods

### 2.1. Literature Search

In this study, we systematically surveyed herbal candidates that showed effects on the liver metastasis of GI-system tumors via the modulation of the TME. Relevant published studies were identified by searching PubMed, Embase, and Cochrane from each database’s inception to December 2021. To search for GI-system tumor metastasis-related articles, we searched for ‘neoplasm metastasis’ OR ‘tumor metastasis’ OR ‘cancer metastasis’ OR ‘metastasis’ OR ‘metast* in the title, abstract, or keywords. In addition, we used ‘herbal medicine’ OR ‘natural’ OR ‘plant’ OR ‘phytogenic’ OR ‘botanical’ OR ‘herb*’ OR ‘phyto*’ OR ‘botanic*’. In order to not to miss results, we manually searched for all cancer metastasis-related articles instead of for any specific organ name or abbreviations. Both free-text searches and MeSH-term searches for titles, abstracts, and keywords were employed. The database searches were supplemented with additional studies from the references of all retrieved relevant studies. The search was restricted to publications in the English language for the collection of the qualified data.

### 2.2. Review and Selection of Relevant Articles

At the time as title and abstract screening, we used the following inclusion and exclusion criteria to select relevant articles. We included human or animal-based studies that focused on the anti-liver metastatic effects of herbal resources, mostly medicinal plants, herbal decoctions, herb-derived compounds, and animal-derived resources. We excluded the following: (i) studies that were not human-derived or animal-based; (ii) studies that included resource(s) already approved for cancer treatment in the National Comprehensive Cancer Network (NCCN) guidelines, such as paclitaxel and irinotecan; (iii) diagnostic method studies; (iv) review articles; (v) studies of metastases in organs other than the liver, such as lung metastases; (vi) retracted articles; and (vii) studies described in other languages. In the full-text screening, we also excluded (i) studies in which the metastatic site was not the liver and (ii) studies in which the described mechanisms were not related to the TME and were only related to the cytotoxic effect of targeting cancer cells (Figure 1).

### 2.3. Analysis of Major Mechanisms Related to the Effects on Metastasis by Targeting the TME

We carefully read the text of the articles and extracted the anti-metastatic mechanisms given by the authors, such as modulation of the extracellular matrix, angiogenesis, endothelial cells, or immunity. In addition, we performed a search for each herbal candidate’s effects on liver environments using hepatopathology terms such as fibrosis, inflammation, injury, and steatosis, which are well-known factors promoting liver metastasis.

## 3. Results

### 3.1. Characteristics of the Selected Studies

Ultimately, 21 studies were included in our analysis. Regarding the origin organs of the cancers, the pancreas was the most frequent (9 studies), followed by the colon (8 studies, including 1 xenograft), the stomach (4 studies), and 1 primary tumor developed using a *Helicobacter felis* infection [3]. A total of 11 studies (57.1%) performed an orthotopic implantation of cell lines in the pancreas, colon, or stomach, while others performed splenic, subcutaneous, intraperitoneal, intravenous, or intrahepatic injections. A total of 12 studies employed an oral administration of herbal candidates, while 7 studies employed intraperitoneal and 2 studies employed intravenous injections (Table 1).

Overall, 15 studies (71.4%) evaluated the efficacy of drugs by comparing the number of metastatic nodules in the liver with those in the control group. Similarly, comparing the size of the liver metastatic area or luciferase activity was also used in some studies. Seven studies compared the effect of their herbal candidates with that of other approved anti-cancer drugs, including FOLFOX (5-FU, fluorouracil, and oxaliplatin), 5-FU, cisplatin, gemcitabine, mithramycin, or capecitabine (Table 1).

### 3.2. Characteristics of the Herbal Candidates

Among the 21 studies, 11 studies researched the effects of 9 herb-derived single compounds, 4 studies researched the effects of 3 single-herb extracts, 4 studies researched the effects of 3 multiherbal decoctions, and 2 studies used an animal-derived resource (1 single compound and 1 water extract). Also included were 3 multiherbal decoctions; *Qing Yi Hua Ji* (consisting of 7 herbs) [4,5], Juzen-taiho-to (10 herbs) [6], and Pien Tze Huang (consisting of 1 herb and 3 materials from animals) [7]. Regarding medicinal plant-derived resources, *Curcuma longa* L. (*C. longa*, including curcumin) was studied in 3 papers [3,8,9], while emodin from *Rheum palmatum* L. [10,11], *Venenum bufonis* (including cinobufacini) [12,13,14], and honokiol from *Magnolia acuminate* L. [14,15] were assessed in 2 studies (Table 2).

### 3.3. Anti-Liver Metastatic Effects on the TME

Regarding the effects on preventing liver metastasis by modulating the TME, 3 studies investigated the TME using metastasized hepatic tissue [7,11,12], while others used the organ/tissues in which the tumor developed (16 studies) or blood (2 studies) (Table 2). In the enrolled studies, the main mechanisms of the herbal candidates could be categorized as effects on cell adhesion or trafficking molecules, angiogenesis, the extracellular matrix, immunity, and inflammatory components.

Eight single compounds, including curcumin, emodin, cinobufacini, and honokiol, increased E-cadherin with/without suppression of the N-cadherin expression in transplanted tumor organs/tissues and/or cells in the circulation [11,13,15,16,17,18,19,20]. The acetate extract of *Celastrus orbiculatus* T. and *Pien tze huang* also had similar effects [7,21]. Water or ethanol extracts of *Curcuma longa* L., *Venenum bufonis*, and *Qing Yi Hua Ji* regulated angiogenesis via the key cytokine vascular endothelial growth factor (VEGF) in metastasized liver or rectum tissues and cancer cells transplanted into the pancreas [4,8,9,12]. Five agents, including emodin and tanshinone II-A, showed anti-liver metastatic effects via the modulation of matrix metalloproteinases (MMPs) with/without effects on the tissue inhibitors of metalloproteinases (TIMPs) [10,12,13,16,19,22]. Two compounds (honokiol and antartina) and two decoctions (*Qing Yi Hua Ji* and *Juzen-taiho-to*) activated tumor immune components, including macrophages and/or dendritic cells [5,6,14,23], while some agents (emodin, honokiol, antartina, *Qing Yi Hua Ji*, and *Pien tze huang*) showed anti-inflammatory effects via the suppression of proinflammatory molecules such as interleukin-6 (IL-6) or nuclear factor-κB (NF-κB) [23] in transplanted tumor tissues, including the colon, pancreas, and peritoneal cavity [5,7,10,18,21]. Further information is summarized in Table 2 and Figure 2.

### 3.4. Effect of Herbal Candidates on the Liver Environment

We additionally investigated the pharmacological effects of these herbal resources on the hepatic microenvironment in 3 liver metastasis-favoring pathologic conditions: hepatic steatosis, inflammation, and fibrosis. As shown in Table 2, 9 of the 15 candidates previously reported beneficial effects against at least one of the three pathologic conditions. Four compounds (curcumin, emodin, honokiol, and tanshinone II-A) and two decoctions (*Juzen-taiho-to* and *Pien tze huang*) previously showed beneficial effects on three pathologic states [26,27,28,29,37,38,41,42]. Deoxyelephantopin showed positive effects against liver inflammation [40], while extracts of both *Annona muricata* L. and *Celastrus orbiculatus* T. showed effects on liver steatosis and inflammation [30,33,34,39]. Further information is summarized in Table 2 and Figure 2.

## 4. Discussion

Medicinal herbs are attractive sources for anti-tumor drug development, and several agents such as paclitaxel, vinblastine, and vincristine are successfully being used owing to their anti-cancer activities [35]. Recently, these natural resources have gained attention not only for cancer prevention and treatment but also for reductions in cancer recurrence and metastasis [36,43]. Given the clinical impact of cancer metastasis, this study systematically reviewed herbal resources with anti-metastatic properties, especially those with effects on the liver metastasis of GI-tract tumors via modulation of the TME.

A total of 21 studies investigated pharmaceutical effects on the liver metastasis of tumors from 3 organs, including the pancreas, colon, and stomach, using 15 types of resources. Four phytochemicals (emodin, honokiol, deoxyelephantopin, and NGD16) and two herbal extracts (*Elephantopus scaber* L. and *Annona muricata* L.), including one decoction (*Qing Yi Hua Ji*), showed anti-liver metastatic effects in four orthotopic implantations and four splenic injection models [5,10,12,14,17,20]. Liver metastasis is a common feature of pancreatic cancer; more than 50% of patients are diagnosed with liver metastases, which is an important poor prognostic factor [44]. However, one clinical study showed that administration with Chinese herbal medicine (CHM) significantly increased the 1-year survival rate in patients with liver-metastasized pancreatic cancer compared with patients who did not receive CHM (21.9% versus 4.8%) [45]. Among the herbal candidates mentioned above, emodin is a main compound derived from *Rheum palmatum* L. that has been traditionally used as a representative herb to treat inflammation-related diseases, including GI-system tumors [46]. Many studies have shown the direct anti-cancer activity of emodin in not only pancreatic tumors but also breast cancer cells [47,48]. From our data, emodin modulated the TME in both tumor-bearing pancreas tissue (suppressing NF-κB and MMP-9) and metastasis-bearing hepatic tissue (increasing E-cadherin) in pancreas and splenic injection models, respectively [10,11].

Regarding gastric and colorectal tumors, the wide and early application of endoscopic screening has led to a reduction in metastasis; however, the death rates due to these malignancies are still very high, ranking second and third, respectively, after lung cancer, especially in many Asian countries [49]. Liver metastasis has the poorest prognostic factor, and it occurs during diagnosis, treatment, or surveillance in approximately 50% of both gastric and colorectal cancer patients [50,51]. Bigelovin, tanshinone II-A, antartina, cinobufacini, curcumin, honokiol, and peperomin E showed anti-metastatic effects via the modulation of MMP-2/9, TIMPs, and N-cadherin under various experimental models of transplanted colon/gastric tumor tissues and cytolytic T cells in blood [3,13,15,16,18,19,23]. The TME-associated activities of these compounds indicate their potential. For example, tanshinone II-A was previously shown to prevent the metastasis of colon tumor cells by the modulation of migration and angiogenesis [52,53]. The ethanol extract of *C. longa* L. and its main compound curcumin have also shown effects on colon/stomach liver metastasis by inhibiting angiogenesis and CXCL12/CXCR4 expression [3,8,9]. *C. longa* L. and curcumin are well-known inhibitors of angiogenesis in a wide variety of tumor cells [54]. The CXCL12/CXCR4 axis plays a crucial role in tumor–stromal cell interactions, leading to progression-related behaviors such as angiogenesis, migration, and metastasis [55].

We reviewed herb-derived agents with actions on the TME. The TME, consisting of cellular and structural components, plays a vital role in the complex and multistep metastatic cascade; thus, the TME has become a target of anti-cancer and anti-metastatic agents [56]. The TME encompasses the surrounding stromal cells, including tumor-infiltrating immune cells, cancer-associated fibroblasts (CAFs), adipose cells, blood endothelial cells, and pericytes as well as the extracellular matrix (ECM), the basement membrane (BM), and cancer-cell secreted growth factors and cytokines such as IL-6 and TGF-β [57]. The herb-derived candidates were found to modulate many of these TME components, mainly inside transplanted tumor tissues, as described above. In our data, two studies (antatina and *Juzen-taiho-to*) revealed the activation of immune cells such monocytes or cytolytic cells in the bloodstream [6,23] and three studies (emodin, *Venenum bufonis*, and *Qing Yi Hua Ji*) showed effects on the hepatic TME of metastases from pancreatic cancer [4,11,12]. Cancer cells encounter immune cells, platelets, and multiple cytokines in the blood microenvironment and need to escape from immune surveillance to successfully metastasize into secondary regions [58]. *Juzen-Taiho-To* showed an effect on the liver metastasis of colon cancer via the regulation of blood immune cells [6].

On the other hand, the state of the liver itself in addition to the primary tumor (stomach, pancreas, or colorectal tumors) also notably affects the end step of metastasis. Alterations in the liver microenvironment such as inflammation, steatosis, and fibrotic changes accelerate liver metastasis [59]. We previously found that the alcohol consumption-induced acceleration of colon–liver metastasis is related to the upregulation of intercellular adhesion molecule 1 (ICAM1) and mild inflammation in hepatic tissue [60]. Furthermore, many studies have found asymptomatic dormant cancer cells in the liver that disseminated from the primary lesion, likely the stomach or colon; under some conditions, the hepatic microenvironment could reactivate cancer cells to proliferate to generate metastasis [61]. One group showed the prevention of colorectal liver metastasis via the administration of interleukin-12 (IL-12), carried in chitosan into the portal vein [62]. IL-12 is one of the most effective inducers of anti-tumor immunity via enhancing the cellular immunity of natural killer cytolytic T cells and interferon (IFN)-γ production [63]. In one cohort study of patients who underwent radical surgery for colon cancer and subsequent routine radiotherapy and chemotherapy, the administration of herbal decoctions showed significant anti-liver metastatic effects, with 26.2 ± 4.5 versus 14.2 ± 4.2 months for the average time to relapse and metastasis, respectively [64]. 

In our study, 9 candidates presented protective or therapeutic effects in pathologically altered hepatic environmental conditions such as inflammation, steatosis, or fibrosis (Table 2). Based on our data, curcumin, emodin, and honokiol are the most promising candidates for inhibiting liver metastasis. These compounds are known to exert pharmacological activities in the TME and liver environment. For example, curcumin inhibits tumor-promoting M2 macrophages [65] and attenuates hepatic inflammation, steatosis, and fibrosis by controlling Kupffer cells and hepatic stellate cells [66]. Both emodin and honokiol have also shown effects on hepatosteatosis, inflammation, and fibrosis in preclinical studies [26,27], along with other effects on TME components such as tumor-specific immune cells [32,67].

The data have some limitations, including the use of only animal studies (no clinical data exist yet) and a relatively small number of articles. It is worth noting that our data collection was limited to English language publications for the collection of the qualified data.

## 5. Conclusions

In this study, we reviewed the recent research findings on herb-derived resources with effects on liver metastasis. Our results provide a basis for scientists working to develop drugs to prevent metastasis via modulation of the TME, especially for GI tumor–liver metastasis.

## Figures and Tables

**Figure 1 cancers-15-03415-f001:**
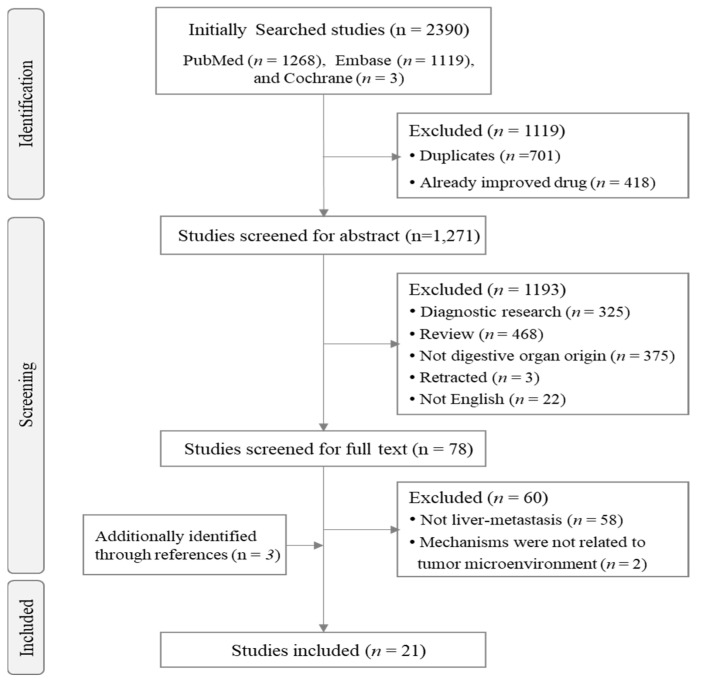
Flow diagram of selection strategy and article review.

**Figure 2 cancers-15-03415-f002:**
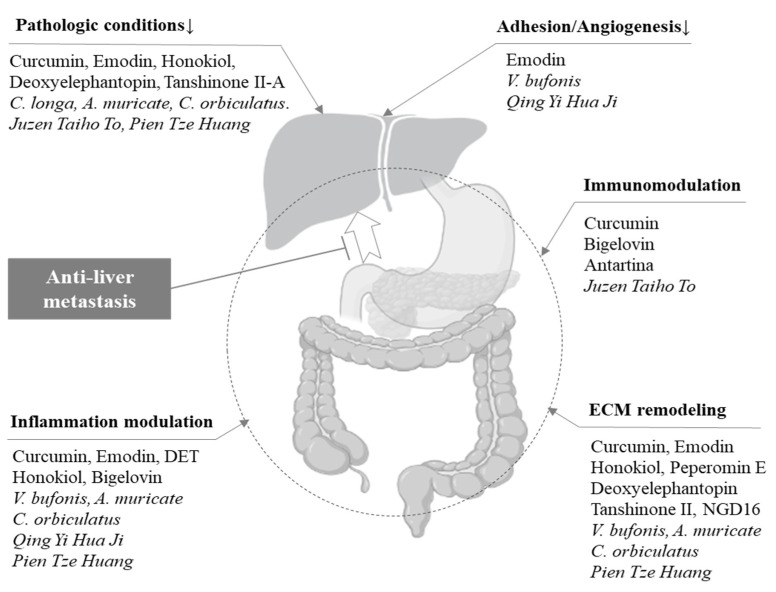
Summary of anti-liver metastatic TMEs’ modulation by herbal candidates. ECM: extracellular matrix; DET: deoxyelephantopin; NGD16: 1,1′-β-D-glucopyranosyl-3,3′-bis (5-bromoindolyl)-octyl methane.

**Table 1 cancers-15-03415-t001:** Summary of the characteristics of the included studies.

Classification	Total Number of Studies/Reference
Final number of included studies	21
Original plant (only single compound or extract)	12
Medication	
Single compound	12/[3,10,11,13,14,15,16,17,18,19,20,23]
Single-herb extract	5/[8,9,12,21,22]
Herbal decoction	4/[4,5,6,7]
Organ of origin	
Pancreas	9/[4,5,10,11,12,14,17,20,22]
Colon	8/[6,7,8,9,13,16,18,23]
Stomach	4/[3,15,19,21]
Experimental design	
Lesion implantation area	
Pancreatic region	4/[10,14,17,22]
Colon region	4/[7,9,13,18]
Stomach region	3/[3,15,19]
Spleen injection	5/[4,5,11,12,15,20]
Subcutaneous injection	1/[8]
Intraperitoneal injection	1/[21]
Intrahepatic injection	1/[23]
Intravenous injection (tail vein)	2/[6,16]
Administration method	
Oral	12/[4,5,6,7,8,9,10,11,16,19,21,22]
Intraperitoneal injection	7/[12,13,14,15,17,20,23]
Intravenous injection	2/[3,18]
Positive control	
FOLFOX (5-FU, fluorouracil, and oxaliplatin)	2/[8,9]
5-FU	1/[23]
Cisplatin	1/[6]
Gemcitabine	1/[20]
Mithramycin	1/[19]
Capecitabine	1/[21]
None	14/[3,4,5,7,10,11,12,13,14,15,16,17,18,22]
Efficacy assessment tool	
Number of liver metastasis nodules	15/[3,4,6,7,10,11,12,13,14,15,16,17,20,21,22]
Size of liver metastatic area	5/[5,8,9,18,23]
Luciferase activity in liver	1/[19]

**Table 2 cancers-15-03415-t002:** List of natural products having preventative effects against liver metastasis of digestive organ cancers.

Original Plant	Origin of Cancer (Method of Transplantation) and Drug Formulation (Route of Administration)	Main Mechanisms Related to the Tumor Microenvironment (Site of Tissue Taken for Assay)	Liver Environment Factors
*Curcuma longa* L.	Xenograft, colon ca. (*sc*), and ethanol ex (*po*) [8]	Regulates EMT and angiogenesis at the original site (rectum)	Steatosis, inflammation, and fibrosis [24,25]
	Colon 26-M01 (*oi*) and ethanol ex (*po*) [9,18]	Regulates immune components and the vascular endothelium at the original site (rectum)	
	Primary gastric cancer cell (*sc*) and curcumin (*iv*) [3]	Inhibits CXCL12/CXCR4 expression at the original site (limb)	
*Rheum palmatum* L.	Pancreas SW1990 (*oi*) and emodin (*po*) [10]	Suppresses NF-κB and MMP-9 protein expression at the original site (pancreas)	Steatosis, inflammation, and fibrosis [26,27]
	Pancreas SW1990 (*si*) and emodin (*po*) [11]	Increases E-cadherin expression via miRNA-1271 at the metastasis site (liver)	
*Venenum bufonis*	Pancreas SW1990 (*si*) and water ex (*ip*) [12]	Regulates MMP-2/9 and VEGF expression at the metastasis site (liver)	
	Colon HCT116 (*oi*) and cinobufacini (*ip*) [13]	Regulates MMP-2/9 and increases E-cadherin expression at the original site (colon)	
*Magnolia acuminate* L.	Pancreas MIA PaCa-2 (*oi*) and honokiol (*ip*) [14]	Interferes with tumor–stromal crosstalk at the original site (pancreas)	Steatosis, inflammation, and fibrosis [28,29]
	Stomach MKN45 (*oi*) and honokiol (*ip*) [15]	Increases E-cadherin and suppresses TGF-β expression at the original site (stomach)	
*Elephantopus scaber* L.	Pancreas BxPC-3 (*si*) and deoxyelephantopin (*ip*) [20]	Increases E-cadherin and suppresses NF-κB expression at the original site (pancreas)	Inflammation [30]
*Inula helianthus-aquatica*	Colon HCT116 (*oi*) and bigelovin (*iv*) [18]	Decreases IL-6 and suppresses N-cadherin expression at the original site (colon)	
*Salvia miltiorrhiza* B.	Colon SW480 (*iv*) and tanshinone II-A (*po*) [16]	Regulates MMP-2/9 and TIMP-1/2 expression at the original site (colon)	Steatosis, inflammation, and fibrosis [31,32]
*Deschampsia Antarctica* D.	Colon CT-26 (*ih*) and antartina (*ip*) [23]	Activates immunity, mainly via dendritic cells and CD8 T cells (blood)	
*Peperomia dindygulensis* M.	Stomach NCI-N87-luc (*oi*) and peperomin E (*po*) [19]	Activates E-cadherin and TIMP-3 at the original site (stomach)	
*Cruciferous vegetables*	Pancreas UN-KC-6141 (*oi*) and NGD16 (*ip*) [17]	Increases E-cadherin expression at the original site in a Par4-dependent manner (pancreas)	
*Annona muricata* L.	Pancreas CD18/HPAF (*oi*) and ethanol ex (*po*) [22]	Suppresses MMP-9 and MUC4 expression at the original site (pancreas)	Steatosis and inflammation [33,34]
*Celastrus orbiculatus* T.	Stomach SGC-7901 (*ip*) and acetate ex (*po*) [21]	Decreases NF-κB, increases E-cadherin, and suppresses N-cadherin expression at the original site (peritoneal)	Steatosis and inflammation [35,36]
*Qing Yi Hua Ji*	Pancreas SW1990HM (*si*) and decoction (*po*) [5]	Decreases IL-6 expression and increases tumor-associated macrophages at the original site (pancreas)	
	Pancreas SW1990HM (*si*) and decoction (*po*) [4]	Suppresses VEGF releases at the metastasis site (liver)	
*Juzen-* *taiho-* *to*	Colon 26-L5 (*pv*) and decoction (*po*) [6]	Regulates macrophages and/or T cells (blood)	Steatosis, inflammation, and fibrosis [37,38]
*Pien* *tze* *huang*	Colon CT-26 (*oi*) and decoction (*po*) [7]	Decreases TGF-β, increases E-cadherin, and suppresses N-cadherin expression at the original site (colon)	Steatosis, inflammation, and fibrosis [39,40]

Ex: extract; DMSO: dimethyl sulfoxide; po: per os (by mouth); sc: subcutaneous injection; oi: orthotopic implantation; iv: intravenous injection; ip: intraperitoneal injection; *si*: splenic injection; *pv: portal vein*; *ih:* intrahepatic inoculation; Wnt: wingless-related integration site; Src: SRC proto-oncogene tyrosine-protein kinase; KRAS: gene Kirsten rat sarcoma viral oncogene homolog; EGFR: epidermal growth factor receptor; SDF-1: stromal cell-derived factor-1; CXCR4: C-X-C chemokine receptor type 4; NF-κB: nuclear factor-κB; miRNA: microRNA; VEGF: vascular endothelial growth factor; SHH: sonic hedgehog; Tpl2: tumor progression locus 2; IL-6: interleukin-6; STAT3: signal transducer and activator of transcription 3; ALK2: activin A receptor type II-like kinase 2; Smad4: mothers against decapentaplegic homolog 4; TGF-β: transforming growth factor-β; NGD16: 1,1′-β-D-glucopyranosyl-3,3′-bis(5-bromoindolyl)-octyl methane: TAM: tumor-associated macrophage; EMT: epithelial mesenchymal transition; MMP: matrix metalloproteinase; TIMP: tissue inhibitor of metalloproteinase.

## Data Availability

The data presented in this study are available on request from the corresponding author.

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
