# Peer review of "A Review of Herbal Resources Inducing Anti-Liver Metastasis Effects in Gastrointestinal Tumors via Modulation of Tumor Microenvironments in Animal Models"

_cancers, 2023, doi:10.3390/cancers15133415_

Round 1

Reviewer 1 Report

1. Line 30 on page 1, the data was too old in 2020. Now is year of 2023. Also at line 68, the date of 2021 is also should be replaced. Related references should be updated.

2. At line 62, why only choose animal-based studies? While at line 84, the human or animal-based studies were included? Inconsistent

3.At line 79, why the search was restricted to publications in the English language? As most research about herbs was published in Chinese. The collected data was incomplete. At last, only 21 studies were included.

 Extensive editing of English language required

Author Response

First of all, I really appreciate reviewer for the professional critique and valuable comments. I have read those comments and have revised my manuscript according to reviewer's comments/suggestions.

1.  Line 30 on page 1, the data was too old in 2020. Now is year of 2023. Also at line 68, the date of 2021 is also should be replaced. Related references should be updated.

=> Many thank reviewer for the important suggestion, I have updated the reference [1].  Regarding the date 'December 2021',  unfortunately we would like to update it later due to the time-limitation of revision due.

2. At line 62, why only choose animal-based studies? While at line 84, the human or animal-based studies were included? Inconsistent

=> We had searched both human and animal studies, while the colleced data were from only animal studies. I have revised the sentence at 'Introduction' section and whole manuscript.

3. At line 79, why the search was restricted to publications in the English language? As most research about herbs was published in Chinese. The collected data was incomplete. At last, only 21 studies were included.

=> I fully understand reviewer's concern. We however decided to restrict only publications in the English language to collect the qualified data.

Reviewer 2 Report

The focus of this review article is to provide an up-to-date review of naturally occurring agents that have a tumor inhibitory effect on liver metastases from primary tumors of the gastrointestinal system. The tumor inhibitory mechanism of such herbal agents is believed to be their effect on the tumor microenvironment (TME). To identify all yet (Dec. 2021) known herbal candidates, the authors perform a systematic review of the online databases PubMed, Embase, and Cochrane. Only animal-based studies are considered in which herbal agents showed an inhibitory effect on liver metastases from tumors of the gastrointestinal system.

A major criticism of this review article is therefore the clear delineation of data between data obtained from patient endpoints and animal studies as stated as the aim of the study.

In line 86 the authors state "We included human or animal-based studies" to be followed in line 87 by "We excluded the following studies: (i) studies that were not animal-based….". This situation (animal-based controled study versus patiend-derived data), which is unclear to the reader, continues throughout the manuscript and needs to be revised. Studies cited in the results or in the discussion that are not based on animal-based data need to be clearly counter-signed.
Overall, however, this review is an interesting and informative compilation of current alternative agents that can provide oncology with innovative therapeutic options, for example in the context of adjuvant therapy options.

A few minor criticisms are listed below.

Line 11:  "…liver metastasis by…. ." The part of a sentence seems to be missing here.

Line 136: Legend of Table 2. There are some abbreviations missing: EMT, MMP, TIMP

Lines 139-140: "…while others used the original organ tissues of cell lines (16 studies) or blood (2 studies)." I do not understand the sense of the sentence. What do they mean by organ tissues of cell lines.

Line 209: "…N-cadherin in under… ." The part of a sentence seems to be missing here.

Lines 229-230: "….immune cells such macrophages or cytolytic cells in the blood stream [6,23],… . " There are no macrophages in the blood, but monocytes. And which cytolytic cells are meant. Here the information of the leucocytic subtypes as described in the paper would be appropriate.

Author Response

A major criticism of this review article is therefore the clear delineation of data between data obtained from patient endpoints and animal studies as stated as the aim of the study.

=> First of all, I really appreciate reviewer for the professional critique and comments. I have read carefully those comments and revised my manuscript according to reviewer's suggestions.

In line 86 the authors state "We included human or animal-based studies" to be followed in line 87 by "We excluded the following studies: (i) studies that were not animal-based….". This situation (animal-based controled study versus patiend-derived data), which is unclear to the reader, continues throughout the manuscript and needs to be revised. Studies cited in the results or in the discussion that are not based on animal-based data need to be clearly counter-signed.

=> I really thank reviewer for the detail and professional comments. The cinclusion criteria are data from human-derived or animal-based studies, although  only animal-based data were included finally due to no data from human study. We have revised the present manuscript consistantly.

Overall, however, this review is an interesting and informative compilation of current alternative agents that can provide oncology with innovative therapeutic options, for example in the context of adjuvant therapy options.

=> I really appreciate reviewer for the positive comments.

A few minor criticisms are listed below.

Line 11:  "…liver metastasis by…. ." The part of a sentence seems to be missing here.

=> Many thank reviewer for the pointing out the mistake, then I have revised the sentence.

Line 136: Legend of Table 2. There are some abbreviations missing: EMT, MMP, TIMP

=> I have revised those abbreviations.

Lines 139-140: "…while others used the original organ tissues of cell lines (16 studies) or blood (2 studies)." I do not understand the sense of the sentence. What do they mean by organ tissues of cell lines.

=>  have revised the sentences.

Line 209: "…N-cadherin in under… ." The part of a sentence seems to be missing here.

=> Many thank reviewer for the correction. I have revised the sentence.

Lines 229-230: "….immune cells such macrophages or cytolytic cells in the blood stream [6,23],… . " There are no macrophages in the blood, but monocytes. And which cytolytic cells are meant. Here the information of the leucocytic subtypes as described in the paper would be appropriate.

=> Many thank reviewer for the detail and professional review and correction. I have revised the sentence.

Reviewer 3 Report

The authors presented a well-written review on “Herbal Resources Inducing Anti-Liver Metastasis Effects in Gastro-intestinal Tumors” with very useful collective tables and figures. But still a few comments need to be addressed.

Introduction

1-      Line 72: you cannot use “we afraid” Please use scientifically sound language.

2-      English editing to the whole manuscript as there are some expressions that need to be improved.

The English language needs to be justified as there are some expressions to be modified. 

Author Response

1.  Line 72: you cannot use “we afraid” Please use scientifically sound language.

=> I really appreciate reviewer for the detail and valuable comment. 

2-      English editing to the whole manuscript as there are some expressions that need to be improved.

=> As reviewer suggested, english edditing has been conducted in the revised manuscript.

Round 2

Reviewer 1 Report

the author had revised the present manuscript according to my former suggestion. besides the following questions.

why only choose english language. related content should be added in the introduction and dicussion part. not just answer it.

no suggestion

Author Response

the author had revised the present manuscript according to my former suggestion. besides the following questions. why only choose english language. related content should be added in the introduction and dicussion part. not just answer it.

=> I really appreciat for the valuble comment and suggestion. I have revised my manuscript about the issue in 'Methods' and 'Dicsussion' sections.
